# The Air–Liquid Interface Reorganizes Membrane Lipids and Enhances the Recruitment of Slc26a3 to Lipid-Rich Domains in Human Colonoid Monolayers

**DOI:** 10.3390/ijms24098273

**Published:** 2023-05-05

**Authors:** C. Ming Tse, Zixin Zhang, Ruxian Lin, Rafiquel Sarker, Mark Donowitz, Varsha Singh

**Affiliations:** 1Division of Gastroenterology & Hepatology, Department of Medicine, School of Medicine, The Johns Hopkins University, 720 Rutland Avenue, 933 Ross Research Building, Baltimore, MD 21205, USA; 2Department of Cellular and Molecular Physiology, School of Medicine, The Johns Hopkins University, Baltimore, MD 21205, USA

**Keywords:** ASMase, DRA, ceramide-rich platform, lipid rafts, SLC26A3, CFTR, colonoids, differentiation

## Abstract

Cholesterol-rich membrane domains, also called lipid rafts (LRs), are specialized membrane domains that provide a platform for intracellular signal transduction. Membrane proteins often cluster in LRs that further aggregate into larger platform-like structures that are enriched in ceramides and are called ceramide-rich platforms (CRPs). The role of CRPs in the regulation of intestinal epithelial functions remains unknown. Down-regulated in adenoma (DRA) is an intestinal Cl^−^/HCO_3_^−^ antiporter that is enriched in LRs. However, little is known regarding the mechanisms involved in the regulation of DRA activity. The air–liquid interface (ALI) was created by removing apical media for a specified number of days; from 12–14 days post-confluency, Caco-2/BBe cells or a colonoid monolayer were grown as submerged cultures. Confocal imaging was used to examine the dimensions of membrane microdomains that contained DRA. DRA expression and activity were enhanced in Caco-2/BBe cells and human colonoids using an ALI culture method. ALI causes an increase in acid sphingomyelinase (ASMase) activity, an enzyme responsible for enhancing ceramide content in the plasma membrane. ALI cultures expressed a larger number of DRA-containing platforms with dimensions >2 µm compared to cells grown as submerged cultures. ASMase inhibitor, desipramine, disrupted CRPs and reduced the ALI-induced increase in DRA expression in the apical membrane. Exposing normal human colonoid monolayers to ALI increased the ASMase activity and enhanced the differentiation of colonoids along with basal and forskolin-stimulated DRA activities. ALI increases DRA activity and expression by increasing ASMase activity and platform formation in Caco-2/BBe cells and by enhancing the differentiation of colonoids.

## 1. Introduction

The intestinal brush-border Cl^−^/HCO_3_^−^ exchanger DRA (SLC26A3) plays a major role in apical Cl^−^/HCO_3_^−^ exchange in the human duodenum, ileum, and colon [1,2,3,4]. DRA takes part in both intestinal Cl^−^ absorption, in which it is linked to Na^+^/H^+^ exchanger 3 (NHE3), and HCO_3_^−^ secretion, where it is linked with CFTR, and functions to reabsorb some of the Cl secreted by CFTR. Plasma membrane trafficking, lipid raft distribution, and protein–protein interactions are the pathways identified, at present, to be involved in the acute regulation of DRA activity in intestinal enterocytes. Previous studies have focused on the mechanisms involved in DRA inhibition occurring in inflammatory diarrheal diseases, including IBDs, caused by enteropathogenic *Escherichia coli* [5,6]. However, very little is known about the factors involved in the activation of DRA activity and the regulation of its secretory function. In polarized enterocytes, DRA is expressed in the apical membrane (but is also associated with tight junctions) and endosomal pools [7,8,9]. In the enterocyte brush border (BB), DRA exists as two distinct populations: one is diffusely distributed and the other is in the lipid rafts (LRs) [7]. Other intestinal BB ion-transport proteins, including NHE3 and the chloride channel CFTR, are also, in part, distributed in BB LRs [10,11,12]. The factors determining the distribution of proteins in the LRs of well-differentiated intestinal epithelial cells are not well-understood. Extensive studies have recently focused on the changes in CFTR activity that are associated with clustering in LRs in pulmonary epithelial cells [13,14]. While the activity of DRA is known to increase with its presence in LRs [7], little is known about the mechanisms regulating DRA localization in LRs, as well as the role of LRs in the basal or regulated activity of DRA.

Lipid rafts are dynamic structures that are assembled from clustered lipids, principally, cholesterol and sphingomyelin (SM), which form a spatially and biochemically distinct domain that segregates specific membrane proteins within the plasma and intracellular membranes. Plasma membrane ceramides are often generated in lipid rafts upon the activation of the enzyme acid sphingomyelinase (ASMase), which cleaves SM into a ceramide and phosphorylcholine (PC) [15,16]. Ceramide converted from SM in lipid rafts exerts its biological activity by altering the structure of lipid rafts [15]. Evidence obtained from studies on airway epithelial cells has suggested ALI as a way to activate ASMase [1,2]. In these studies, epithelial cells were typically seeded on the upper surface of semi-permeable membrane inserts, and the cells were fed, basolaterally, with a culture medium. The contact with air on one side and nutrients on the other imposed polarity and partially mimicked the in vivo conditions of mature adult airways.

In the current study, we demonstrate the presence of raft-like clusters of DRA in the apical domain of Caco-2/BBE cells that are closely associated with acid sphingomyelinase (ASMase). In basal conditions, ASMase was bound to the extracellular surface of the BB of Caco-2/BBe cells. In the ALI cultures, the ASMase was activated and present in aggregates along with aggregated DRA on the plasma membrane, including in detergent-insoluble fractions. These were associated with enhanced DRA expression and increased basal and forskolin-stimulated DRA activities. Moreover, the ASMase inhibitor, desipramine, decreased both basal ASMase quantity and activity, and the increases caused by ALI along with a decrease inDRA basal expression as well as an ALI-induced increase in the BB amount of Flag-DRA. The enhanced formation of platform-like structures is a well-established signaling mechanism under a wide range of pathological conditions [4,14]. The present results indicate that these platform-like structures also contribute to the activation of DRA in intestinal epithelial cells. The acute increase in DRA activity due to the formation of BB ASMase-dependent platform-like structures is a newly defined mechanism of DRA regulation.

## 2. Results

### 2.1. ALI Culture Increases DRA Expression in the Apical Membrane in Caco-2/BBe Cells

To study the effect of ALI exposure, apical media were removed from 12–14 days post-confluency, Caco-2/BBe cells grown as submerged cultures and were studied 2 days later (Figure 1A). The 2-day ALI-exposed cells were significantly taller compared to the cells in the submerged culture (submerged h = 12.0 ± 1.3 µm; ALI h = 22.0 ± 1.8 µm; *p* < 0.05), while the transepithelial electrical resistance (TEER) was not significantly affected by the ALI culture (Figure 1B,C). In the immunostained Transwell inserts of Caco-2/BBe cells grown in the submerged culture, DRA was primarily localized at the apical surface and, under baseline conditions, was generally distributed as small punctate structures (white arrowheads), while, in some areas, there were DRA aggregates (red arrowheads) (Figure 1D). The sensitivity of the DRA antibody used in this study was reported in our previous studies [17,18]. The exposure of confluent Caco-2/BBe cells to ALI for 2 days caused a major reorganization of DRA, which was present in compact aggregates at the apical membrane in our study. Multiple, large aggregates of apical DRA are shown with red arrowheads in the orthogonal view, XZ section, and XY section close to the top of the BB (Figure 1D). ALI was associated with a threefold increase in DRA expression (3.5 ± 0.08-fold, both bands together) compared to cells grown in submerged cultures (Figure 1E). The DRA higher molecular weight highly glycosylated band was significantly increased with ALI 4.9 ± 0.5-fold) (Figure 1E), while the core glycosylated band was not significantly altered (1.5 ± 2.2-fold) when compared with the cells grown submerged throughout the experiment [17,18]. Importantly, the increase in DRA protein expression due to ALI modification was significantly greater than the change in DRA mRNA (Figure 1F). Caco-2/BBe DRA mRNA expression was increased 1.53 ± 0.3-fold in ALI culture compared to the cells grown in the submerged culture. Caco-2/BBe cells exposed to 4 and 6 days of ALI also had increased expressions of DRA (Appendix A). However, Caco-2/BBe cells became multi-layered when exposed to the ALI culture for more than 2 days (Appendix A). Therefore, the rest of the studies were performed on monolayers of Caco-2/BBe cells with 2 days of ALI. 

To investigate the alterations in other ion-transport proteins after ALI, the mRNA levels of selected ion transporters and the BB protein sucrase-isomaltase (SI) were also compared between submerged and ALI cultures. The ion transporters selected are known to play important roles in Cl^−^ and HCO_3_^−^ secretion, electroneutral Na^+^ absorption, and intracellular pH regulation. As shown in Figure 1F, several ion transporters are significantly upregulated at the mRNA level in Caco-2/BBe cells after 2-day ALI. These include NHE1 (2.2-fold), electroneutral Na^+^/HCO_3_^−^ co-transporter 1 (NBCe1) (2.4-fold), and putative anion transporter 1 (PAT-1) (1.7-fold). Sucrase-isomaltase (SI), a brush-border glucosidase and intestinal differentiation marker, was also significantly upregulated at the mRNA level in ALI cells. Some ion transporters were significantly downregulated after ALI, including the potassium voltage-gated channel, subfamily E, regulatory subunit 3 (KCNE3) (2-fold). The mRNA levels of the following ion transporters were not significantly changed by ALI: Anoctamin-1 (ANO1), potassium voltage-gated channels, subfamily Q, member 1 (KCNQ1), NKCC1, and CFTR (Figure 1F). Overall, these results suggest that DRA resides at the apical side of enterocytes and that DRA undergoes extensive reorganization in the apical membrane, along with an increase in expression during ALI culture, which includes both transcriptional changes as well as post-translational modifications. These changes are not specific to DRA and ALI affects the expression of multiple apical and basolateral transport proteins and at least one BB digestive enzyme.

### 2.2. ALI Culture Causes an Increase in ASMase Activity and Colocalization with DRA

Aggregates, such as those of DRA in ALI-exposed Caco-2/BBe cells, have been previously reported to occur due to the increase in ceramide content in the plasma membrane [14]. Ceramide changes the biophysical properties of the membrane and causes lipid rafts to cluster together [15,19]. ASMase is an enzyme that causes an increase in ceramide content in the plasma membrane by cleaving phosphorylcholine from sphingomyelin [20] (Figure 2A). The translocation of ASMase in response to stimuli was previously identified using the immunostaining of ASMase under nonpermeabilized conditions [13]. To examine if ASMase increases with ALI and if it is translocated to the BB during ALI culture, we compared ASMase immunostaining using confocal microscopy to examine the apical domain of unpermeabilized, well-differentiated Caco-2/BBe-Flag DRA cells grown submerged and after ALI exposure. ASMase immunofluorescence was readily detected in non-permeabilized Caco-2/BBe cells in both conditions, while increased immunofluorescence was detected after ALI stimulation. By performing immunostaining, Flag-DRA and ASMase were co-localized under both submerged and ALI conditions. Moreover, the increased localization of Flag-DRA and ASMase was observed in the apical domain (shown in white, enclosed areas) after two days of ALI exposure to Caco-2/BBe cells (Figure 2B). Manders’ overlap coefficient analysis showed a 90% spatial colocalization between Flag-DRA and ASMase at the BB (Figure 2B). The BB expression of DRA was compared between submerged and ALI-exposed Caco-2/BBe cells using cell surface biotinylation. The analysis of the surface-to-total ratio of DRA was significantly increased after ALI exposure compared to the submerged culture (Figure 2C). Since CFTR surface expression was previously reported to be elevated due to an increase in membrane ceramide levels in airway epithelial cells [14], we also compared the CFTR surface expression between submerged and ALI cultures in Caco-2/BBe cells. Similar to DRA, the surface expression of CFTR was significantly increased in Caco-2/BBe cells after 2 days of ALI exposure (Figure 2C). To determine if ASMase activity was stimulated by ALI, we used a fluorometric assay, as described in the Materials and Methods Section. ASMase activity increased following ALI exposure (submerged: 1.5 ± 0.5; ALI: 3.5 ± 0.4; *p* < 0.05 vs. submerged) (Figure 2D). These results indicate that ASMase is constitutively present on the exterior surface of the BB of Caco-2/BBe cells, and its amount and activity increase with ALI exposure, similar to the increase in DRA localization at the BB.

### 2.3. Basal and Agonist-Stimulated DRA Activity in Intestinal Epithelial Cells Is Increased by Exposure to ALI Culture

Basal DRA activity, measured by the initial rate of alkalinization that immediately follows the removal of Cl^−^ from the apical HCO_3_^−^ bathing solution was compared between Caco-2/BBe cells grown as submerged culture or modified as ALI for 2 days. The initial rate of Cl ^−^/HCO_3_^–^ exchange was significantly increased by ALI (135.0 ± 1.5%) (Figure 3A,B). We recently showed that forskolin acutely stimulates DRA activity, which, in part, involves trafficking to the apical plasma membrane [17]. Therefore, we compared the responses of forskolin between the submerged and ALI-modified Caco-2/BBe cells. Forskolin (10 μmol/L, 10 min) significantly increased the DRA activity in submerged Caco-2/BBe (127.0 ± 8.1%) cells and caused a non-significantly greater increase in ALI-modified cells (154.0 ± 0.5%). We then investigated if forskolin caused a fusion of DRA into clusters that looked similar to the demonstrated ceramide-rich platforms in human bronchial epithelial cells. Initially, the dimensions of the membrane microdomains that contain DRA were examined and the number of DRA clusters per unit area prior to and following forskolin stimulations was compared. Based on the previous validations and nomenclature, protein-containing membrane areas presenting dimensions close to the limit of the optical resolution (<0.25 µm diameter) were as considered clusters, whereas large aggregates (≥2 µm dia.) were considered as platforms [13,14]. Quantitation in Caco-2/BBe cells showed that the cells grown in a submerged culture had higher numbers of DRA clusters (<0.25 µm) (average size: 1.2 ± 0.3 µm), compared to 2-day ALI cultures (average size: 4.8 ± 0.3 µm), but presented a much lower number of platforms (≥2 µm) (Figure 3C,D). In the submerged cultures, forskolin resulted in an increase in the number of platforms (average size: 3.0 ± 0.4 µm) and a significant decrease in the number of clusters. In contrast, in Caco-2/BBe cells, following ALI exposure, DRA was only present in the platforms (≥2 µm dia), and the size of the platforms increased further (average size: 6.1 ± 0.4 µm) in response to the forskolin (Figure 3D). We also compared the Fsk-induced short-circuit increase between the submerged and ALI cultures in Caco-2/BBe cells. Similar to the reports on primary human bronchial epithelial cells, there was a significant increase in forskolin-stimulated CFTR activity (Figure 3E) following 2 days of ALI exposure. These results suggest that an increase in the size of DRA-containing clusters in the plasma membrane to form platforms due to ALI may contribute to the increase in basal and forskolin-stimulated DRA activities in these cells.

### 2.4. ASMase-Inhibitor Desipramine Treatment Reduces ASMase and Basal DRA Expressions as Well as the ALI-Induced Increase in DRA Quantity at the BB

DRA is, in part, localized in the lipid raft pools of the Caco-2/BBe plasma membrane [7,21]. Since ASMase and DRA co-localize and are present in increased amounts in apical membrane platforms in ALI, the dependence of DRA abundance in the plasma membrane on ASMase was determined. Desipramine, which functionally inhibits ASMase, was also used [22]. Caco-2/BBe cells were transduced with adenoviral Flag-DRA; then, 24 h later, the cells were exposed to desipramine for 1 h (13 µM), followed by exposure to ALI for 2 days in the continued presence of desipramine (13 µM). As shown in Figure 4A,B, desipramine decreased both the basal ASMase amount and activity and the increases caused by ALI. Desipramine pretreatment also decreased the basal expression as well as the ALI-induced increase in the BB amount of Flag-DRA (Figure 4A). Importantly, pretreating cells with desipramine for 1 h reduced ASMase activity to below the control level, suggesting that there was ASMase activity in the Caco-2/BBe cells under basal conditions. The results indicate that the ALI-induced accumulation of DRA in the apical membrane depends on the activity of ASMase.

### 2.5. ALI Causes an Increase in the Detergent-Insoluble Fraction (Lipid Raft) of DRA in Intestinal Epithelial Cells

Lipid rafts are enriched with cholesterol and sphingolipids and are resistant to detergent solubilization, such as by Triton X-100; hence, they are included in the detergent-insoluble (DI) membrane fraction [7]. Since insolubility depends on the cholesterol composition of these microdomains, we determined the effect of methyl-β-cyclodextrin (MβCD) (selectively removes most of the cholesterol from the membrane) on the association of DRA with the DI fraction. Total membrane from DMSO or MβCD (10 µM)-treated submerged Caco-2/BBE cells were incubated with Triton-X 100 at 4 °C and the DI and DS fractions were separated and analyzed using Western blot analysis. As shown in Figure 5A, the DI fraction of the total membrane prepared from Caco-2/BBe cells expresses more DRA compared to the DS fraction under basal conditions. Moreover, the treatment with MβCD caused a significant reduction in the expression of DRA in the DI membrane fraction (Figure 5A). These findings support the previous reports of the presence of DRA in lipid rafts in the total membrane of Caco-2/BBe cells [7,21]. Whether ALI altered the association of DRA with lipid rafts or the DI fraction of the Caco-2/BBe total membrane was investigated. As shown in Figure 5B, a higher amount of DRA was expressed in the DI compared to the DS fraction of both the submerged (control) and ALI-exposed Caco-2/BBe cells. However, DRA expression in the DI fraction was significantly enhanced in the ALI compared with the submerged culture. These findings suggest that ALI increases the amount of DRA in the lipid raft fraction of the plasma membrane.

### 2.6. ALI Enhances the Differentiation of Human Colonoids

Caco-2/BBe cells are human colon cancer-derived cell lines. To determine whether ALI also affects DRA in normal human intestinal epithelial cells, we used colon-derived normal human intestinal enteroids, also called colonoids (HIEs). When HIEs are differentiated, they include the different epithelial cell types present in the normal intestine, and studies were performed in monolayers of differentiated HIEs because DRA is expressed almost entirely in this population [17,18,23]. The monolayers were initially maintained in the undifferentiated (UD) crypt-like state by growth in Wnt3A, Rspo-1, and Noggin, and for these studies were caused to differentiate (DF) by the withdrawal of growth factors (Wnt3A and Rspo-1) for 5–6 days [23,24]. DRA expression was compared under four different conditions: (1) UD, (2) UD + ALI—5 days, (3) DF, and (4) DF + ALI—5 days. UD enteroid monolayers from submerged or ALI conditions did not exhibit a significant time-dependent increase in TEER. In contrast, the TEER of the differentiating monolayers from both the submerged and ALI conditions increased significantly and similarly over time (Figure 6A). In these studies, even after 5 days of ALI exposure, colonoid monolayers did not form multilayers, as observed in the Caco-2/BBe cells (Appendix A).

To investigate if ALI altered the differentiation of the colonoids, the quantitative RT-PCR expression of a selected panel of genes was performed on monolayers collected from the abovementioned four conditions. As shown in Figure 6B, *Lgr5* and *Ki67* mRNA reduced in all conditions, except the UD enteroids. Similarly, atonal homolog 1 (*ATOH1*) and mucin 2 (*MUC2*) expressions increased in all conditions, compared with UD monolayers. Additional markers of enterocyte differentiation were examined. Sucrase-isomaltase and alkaline phosphatase increased similarly in all conditions, except in the UD monolayers [25]. Further support for the effect of ALI on enteroid differentiation is demonstrated in Figure 6C, in which the mucus layer of the enteroids was determined by confocal microscopy. ALI increased the thickness of the mucus layer in the UD + ALI condition and DF + ALI monolayers with the largest mucus layer present in the DF + ALI condition.

DRA mRNA expression was significantly increased in UD + ALI for 5 days (~2.5-fold vs. UD) and DF (~15-fold vs. UD) monolayers. However, cells from DF + ALI-5days monolayers showed a very high upregulation of DRA mRNA expression (~60-fold vs. UD). The DRA protein expression was also measured using the Western blot analysis of lysates prepared from the colonoids (Figure 6D). The DRA protein expression tracked the changes in mRNA as being significantly increased by DF and DF + ALI conditions, with the greatest change in the DF + ALI condition, an increase compared to the 18-fold UD, while the DRA expression in the DF condition was increased 5-fold vs. UD. UD + ALI caused a slight, but not significant, increase in DRA protein expression. In addition, the apical membrane expression of DRA was increased in all three conditions compared to the UD enteroids, with the greatest increase being in the DF + ALI conditions (Figure 6E). The ASMase activity in cells under the abovementioned four conditions was also determined (Figure 6F). Compared to UD monolayers, which had very low ASMase activity, monolayers from the other three conditions had higher ASMase activities; however, these were only statistically significant for the DF and DF + ALI conditions, with the highest ASMase activity in the DF+ ALI condition (Figure 6F). Overall, these results suggest that exposure of monolayers to ALI enhances the differentiation of colonoids, which is associated with an increase in DRA and ASMase expressions. The increase in differentiation occurred much more frequently in monolayers in which differentiation was initiated by growth factor removal, and the changes were much less frequent in undifferentiated enteroids.

### 2.7. DRA Expression and Activity Are Further Enhanced by ALI Culture of Differentiated Human Colonoids

To explore the role of ALI in enhancing the differentiation of already differentiated human colonoids, we exposed enteroids differentiated for 4 days to 2 subsequent days of ALI and studied the changes in the specific gene expressions, including DRA. These studies were performed in the continued presence of DF media on the basolateral side throughout. The changes in DRA expression were compared between colonoids from 4-day DF + 2-day ALI with 6-day DF. The TEER was not significantly different between 6-day DF and the 4-day DF + 2-day ALI culture (Figure 7A). In contrast, the mRNA expression of differentiation markers, including sucrase-isomaltase and alkaline phosphatase, increased further, along with DRA, which present a small, but significant increase, in the ALI condition (~1.4 folds vs. 6 day-DF) as compared to 6-day DF (Figure 7B). Of note, the protein expression of DRA was ~3-fold higher in the 4-day DF + 2-day ALI culture as compared to the 6- day DF culture (Figure 7C). Accordingly, basal DRA activity was ~1.6-fold higher in the 4-day DF + 2-day ALI culture (Figure 7D). We also compared the stimulation of forskolin on monolayers under these two conditions. The percentage of stimulation of DRA in response to forskolin was not significantly different between the 4-day DF + 2-day ALI culture and 6-day DF culture (DF:150± 0.3%; DF + ALI: 137± 0.1%; *p* = n.s vs. 6-day DF) (Figure 7E). The immunofluorescence analysis of monolayers also showed a higher expression of DRA on the apical membrane and larger DRA-containing aggregates in the 4-day DF + 2-day ALI culture (Figure 7F). Moreover, significantly higher ASMase activity was detected in cells from the 4-day DF + 2-day ALI culture as compared to the 6-day DF, and both were significantly higher than the ASMase activity in UD monolayers (Figure 7G). These results show that the DF of colonoids increases the ASMase activity as well as DRA expression, and that the exposure of DF colonoids to ALI further increases ASMase activity along with DRA expression.

## 3. Discussion

In this study, we showed that ALI exposure enhances the differentiation of human proximal colonic enteroids and Caco-2 cells; this occurred in previously differentiated enteroids as well as when cells were exposed to ALI during growth factor-removal-induced differentiation and, although to a lesser extent, in undifferentiated enteroids. The aspects of differentiation that were affected included the reduced expression of stem cell markers, including the mRNA expression of *LGR5*, reduced proliferation (*Ki6*7), increased expression of *ATOH1*, along with the goblet cell marker *MUC2*, and a thicker apical mucus layer. Additionally, there was an increased expression of apical markers in villus cells (sucrase-isomaltase and alkaline phosphatase), as well as DRA, which is normally present only in enterocytes that are at least partially differentiated, and greatly increases in quantity with differentiation. Similar changes also occurred in UD enteroids when exposed to ALI; although, the extent of the increase was reduced and often failed to reach statistical significance. These changes included significantly reduced mRNA for *LGR5*, and increased *ATOH*1 and *MUC2* expressions along with a thicker colonic mucus layer. Our previous findings in human duodenal enteroids suggested that differentiated (villus-like) enteroids express more DRA, NBCe1, and sucrase-isomaltase, and reduced KCNE3 and NKCC1 expressions [18]; all changes were similar to the effects caused by ALI in the current study. Together, these findings show that ALI enhances the differentiation of human colonoids. ALI is a standard method for the differentiation of human bronchial epithelial cells, and it was recently shown that ALI-like effects can be achieved in submerged culture by controlling the oxygen content [26,27]. The latter findings suggest that oxygen levels play a significant role in the differentiation of epithelial cells. While the effects of oxygen levels on enteroid differentiation have not been directly studied, our current findings are supported by previous studies suggesting that the long-term culturing of organoids as submerged monolayers generates low oxygen tension, leading to cellular stress, reduced differentiation, and a lack of secretory cells. Notably, these effects were reversed by the ALI culture that increases the oxygen levels of the enteroid monolayers and re-establishes healthy and differentiated monolayers [28,29,30].

In this study, we emphasized two proteins that underwent increased expression as part of ALI-induced differentiation: ASMase and DRA. ASMase is an enzyme that breaks sphingomyelin into ceramide and phosphorylcholine in the outer leaflet of the plasma membrane. Ceramide changes the biophysical properties of membranes and causes lipid rafts to cluster together to form ceramide-rich platforms (CRPs), previously observed in airway epithelial cells, lymphocytes, and fibroblasts [14,31]. The in vitro studies revealed that the generation of ceramide is sufficient to trigger the formation of distinct platforms, even in purely artificial membranes without any cytoskeleton or other cellular proteins [32]. These platforms selectively trap or exclude specific proteins for biophysical and energetic reasons, and thus serve as a sorting unit for receptors and signaling molecules [31]. ALI increased ASMase activity in human colonoids and Caco-2/BBE cells and visibly enlarged apical membrane ASMase-containing aggregates that fit the description of ceramide-rich platforms [14,33]. ASMase is ubiquitously expressed and localizes preferentially to the endo-lysosomal compartment. However, under certain conditions, it translocates to the plasma membrane [34]. In the ALI of Caco-2/BBe cells, the plasma membrane translocation that occurred was demonstrated based on the ability to observe its presence on the apical surface in non-permeabilized cells. Under basal conditions, ASMase co-localized with DRA in apical membrane clusters, and with ALI, these clusters aggregated into CRPs that contained DRA. How the ALI-induced activation of ASMase models a normal physiologic or pathologic regulation of ASMase has not been established. ASMase is considered to have a protective role during infection, based on the studies on ASMase knockout mice, which had exaggerated IL-1β release and increased mortality when infected with *Pseudomonas aeruginosa* [35]. ASMase is activated in response to various stress stimuli, including (1) the ligation of death receptors (tumor necrosis factor-α, CD95, and TRAIL); (2) radiation (UV-C and ionizing radiation); (3) chemotherapeutic agents (cisplatin, doxorubicin, paclitaxel, and histone deacetylase inhibitors); (4) viral, bacterial, and parasitic pathogens (*rhinoviruses, Neisseria gonorrhea*, *Staphylococcus aureus*, *Pseudomonas aeruginosa,* and *Cryptosporidium parvum*); and (5) cytokines (e.g., IL-1β) [36,37,38,39,40]. Evidence from these studies in cell culture and animal models has been interpreted to suggest that the activation of ASMase takes part in regulating cellular differentiation (monocyte to macrophages), growth arrest, apoptosis, and immune defense mechanisms [41,42]. However, its possible role in stimulating enterocyte differentiation has not been suggested. Changes in enterocyte differentiation occur in multiple models of intestinal pathology, including infection by bacteria, viruses, and worms, as well conditions of chronic inflammation. Given that DRA activity has been altered in several of these models, it is worth determining whether ASMase activation/plasma membrane translocation has a pathophysiologic role, and given its presence under basal conditions in the Caco-2/BBe plasma membrane, its role in regulating basal DRA activity under normal physiologic conditions that include the post-prandial state is important to understand as well.

Parallel to the changes in ASMase, 2 days of ALI significantly increased the amount of total and apical membrane DRA and increased the localization of DRA in the detergent-insoluble portion of the total and apical membranes. The increased DRA was almost entirely in the upper, heavily glycosylated band with minimal changes in the core glycosylated band, consistent with effects primarily on the apical membrane DRA pool [17]. These results are consistent with the previous reports on Caco-2/BBe cells that showed that the majority of DRA is present in the detergent-insoluble membrane fraction [7]. An increase in DRA expression in human colonoids after a 6-day ALI culture was primarily due to the transcriptional upregulation of the DRA gene expression due to enhanced differentiation of colonoids. However, when 4-day DF colonoids were exposed to 2 days ALI they showed enhanced DRA protein expression and activity, with less transcriptional upregulation. This effect was similar to Caco-2/BBe cells where ALI showed a greater effect on translational upregulation than via transcriptional stimulation (Figure 1C). In airway cells, the enhancement of CRPs during regulation by physiological agonists is known to increase CFTR activity by triggering membrane lipid conversion into ceramide and the fusion of ceramide-rich platforms [13]. Similarly, in Caco2-BBe cells, we found a significant increase in CFTR apical expression (Figure 2C) and activity after 2 days of ALI exposure (Figure 3E). This suggests that there is a common mechanism of activation of CFTR by these platforms in both the pulmonary and intestinal cells, as well as a similar mechanism by which ALI affects DRA and CFTR in human enteroids. We recently reported that forskolin stimulation of DRA is at least, in part, dependent on CFTR protein expression, supporting the studies of Ko and Muallem on oocytes [17,43]. It remains to be determined if CRPs have a role in the stimulation of DRA activity in response to forskolin, in addition to the potential increase in the accessibility that results from having both DRA and CFTR present in CRPs. Further studies are required to understand the regulation of DRA under physiologic and pathologic conditions that are associated with increased amounts of CRP.

In summary, the present results for ALI demonstrate that it causes enterocyte differentiation. Concerning the relevance of these findings to normal physiology, the intestine is primarily exposed to high luminal volumes related to eating, while much less luminal fluid is present during many hours of fasting or sleep. In this study, we identified a new pathway for DRA stimulation that extends our understanding of apical compartmentalized DRA and potentially provides insights into how it interacts with CFTR. The physiologic and pathophysiologic significances and factors involved in the increase in ASMase activity in the plasma membrane remain to be established; although, these results strongly support a role of ASMase activation during DRA and CFTR regulation processes. Moreover, understanding the distribution and lateral mobility of DRA in ceramide-rich platforms in the plasma membrane and how these aspects of DRA function, which are regulated by secretagogues, will provide insights into better understanding the role of CRPs in the regulation of multiple brush-border ion-transport processes.

## 4. Materials and Methods

Chemicals and reagents were purchased from Thermo Fisher (Waltham, MA, USA) or Sigma-Aldrich (St. Louis, MO, USA) unless otherwise specified.

### 4.1. Cell Lines

Caco-2/BBe cells were cultured in Dulbecco’s modified Eagle medium supplemented with 25 mmol/L NaHCO_3_, 0.1 mmol/L nonessential amino acids, 10% fetal bovine serum, 4 mmol/L glutamine, 100 U/mL penicillin, and 100 μg/mL streptomycin in a 5% CO_2_/95% air atmosphere at 37 °C. For experiments, cells were plated on Transwell inserts (Corning, Inc., Corning, NY, USA) and studied 14–18 days after reaching confluency. The plasmid p3xFLAG-DRA was cloned into the adenoviral shuttle vector ADLOX.HTM under the control of a cytomegalovirus (CMV) promoter [18].

Endoscopic specimens of the human proximal colon from healthy human subjects undergoing endoscopies for medically indicated conditions were used to establish primary cultures of the human colon, called colonoids, as previously described [44]. Colonoids were expanded and plated on Transwell inserts (polyester membrane with 0.4-μm pores Corning) to form monolayers, as previously described [17,18,24,44]. The formation of colonoid monolayers was monitored by the measurement of TEER. Monolayers were maintained in an undifferentiated state by exposure to WNT3A, Rspon-1 and Noggin [24]. For differentiation, colonoids were maintained in a medium that lacked WNT3A, R-spondin1, and SB202190 for 5 days. Five days later, paired UD and DF enteroid monolayers were studied. Most results of the current study were obtained from colonoids derived from 1 healthy donor, with similar results observed in colonoids from 2 other donors. The procurement and study of human colonoids were approved by the Institutional Review Board of Johns Hopkins University School of Medicine (NA_00038329).

For air-liquid interface studies, apical culture media were removed for specified days from 12–14 days post confluent Caco2/BBe cells or confluent colonoid monolayers.

### 4.2. Immunofluorescence

Cells were grown on collagen-coated Transwell supports as submerged culture or exposed to an air-liquid interface for a specified number of days. To visualize DRA and the effects of ALI on DRA, cells were washed with PBS and fixed in 4% paraformaldehyde for 30–45 min, incubated with 5% bovine serum albumin/0.1% saponin in phosphate-buffered saline for 1 h, and incubated with primary antibody against Flag or DRA (mouse monoclonal, 1:100, sc-376187; Santa Cruz, Dallas, TX, USA) alone or in combination with a rabbit anti-ASMase antibody (1:200; Abcam ab227966) overnight at 4 °C. Cells were then exposed to Alexa Fluor 488 goat anti-mouse and Alexa Fluor 594 goat anti-rabbit secondary antibodies (1:1000 dilution; Invitrogen, Waltham, MA, USA) for 1 h and mounted in ProLong Diamond Antifade Mountant (Invitrogen, Waltham, MA, USA). Images were collected using ×40 oil immersion objective on an FV3000 confocal microscope (Olympus, Tokyo, Japan) with Olympus FV31S-SW and Fiji (ImageJ-2020) (NIH). For quantitative analysis, the same settings were used to image all samples. To examine ASMase and DRA distribution on the apical surface of Caco-2/BBE cells, the above procedure was followed without the membrane permeabilization step. The effect of ASMase inhibitor desipramine on ASMase and DRA distribution was assessed by pretreating cells on both sides of Transwells for 1 h and then exposing them to 2 days-ALI in presence of desipramine. Individual clusters (optical resolution (<0.25 µm)) and platforms (large aggregates (≥2 µm dia.)) were encircled using Fiji (ImageJ-2020), and their areas were estimated from the number of pixels × single pixel area [13,14]. Cluster size was quantified using pixel number and total area. Microdomain areas were estimated by counting three areas/monolayers/conditions from three different experiments.

Analysis of MUC2 by immunofluorescence and confocal microscopy was carried out as previously reported [44,45,46]. Briefly, human colonoid monolayers were fixed with Carnoy’s solution (90% [*v*/*v*] methanol, 10% [*v*/*v*] glacial acetic acid), washed 3 times with phosphate-buffered saline, permeabilized with 0.1% saponin, and blocked with 2% bovine serum albumin + 15% fetal bovine serum for 60 min (all Sigma-Aldrich), followed by overnight incubation with antibody and secondary antibody staining. Images were collected using ×40 (NA 1.25) oil immersion objectives on FV3000 confocal microscope (Olympus, Tokyo, Japan) with Olympus FV31S-SW and Fiji (ImageJ-2020) (NIH). Images were 3D-reconstructed using Volocity Image Analysis software (v6.1, PerkinElmer; USA). Primary antibody mouse anti-MUC2 (Santa Cruz Biotechnology, Dallas, TX, USA; sc7314) at 1:100 dilutions was used.

### 4.3. ASMase Activity

The enzymatic hydrolysis of SM to ceramide and phosphorylcholine by acid sphingomyelinase was measured with the Acid Sphingomyelinase Activity Assay Kit (Echelon Biosciences, K-3200) (Salt Lake City, UT, USA) according to the manufacturer’s instructions. Caco2/BBe cells and colonoids grown in Transwell inserts were washed with cold PBS and then scraped in 1 mM PMSF on ice. Subsequently, cells were sonicated in an ice water bath sonicator for 10 min and then were disrupted by three rounds of freeze–thaw cycles with liquid nitrogen with vortexing in between each cycle. Extracts were clarified by centrifugation for 12 min at 16,000× *g* at 4 °C, and protein concentrations were determined by the BCA protein assay (Thermo Scientific, Waltham, MA, USA). The enzymatic assay was carried out in 96-well plates. Each well contained 5 μg of protein in 20 μL of 1 mM PMSF. ASM standards were prepared in 1 mM PMSF as well. Substrate buffer (Echelon Biosciences, K-3203) (Salt Lake City, UT, USA), 30 μL, was added to each well containing 20 μL of sample or standard. Then, 50 μL/well of the diluted substrate (Echelon Biosciences, K-3202) in substrate buffer was added to each well. The plate was incubated for 3 h at 37 °C with agitation. Stop buffer (Echelon Biosciences, K3204) was added to each well, and fluorescence was measured after 10 min of shaking at RT using FLUOstar Omega plate reader at 360 nm excitation and 460 nm emission. Data are presented as pmol substrate degraded/μg protein/min.

### 4.4. Immunoblotting

Cells were rinsed 3 times and harvested in phosphate-buffered saline by scraping. Cell pellets were collected by centrifugation, solubilized in lysis buffer (60 mmol/L HEPES, 150 mmol/L NaCl, 3 mmol/L KCl, 5 mmol/L EDTA trisodium, 3 mmol/L ethylene glycol-bis(β-aminoethyl ether)—*N*,*N*,*N′*,*N′*—tetraacetic acid, 1 mmol/L Na _3_ PO _4_, and 1% Triton X-100, pH 7.4) containing a protease inhibitor cocktail, and homogenized by sonication. Protein concentration was measured using the bicinchoninic acid method. Proteins were incubated with sodium dodecyl sulfate buffer (5 mmol/L Tris-HCl, 1% sodium dodecyl sulfate, 10% glycerol, 1% 2-mercaptoethanol, pH 6.8) at 37 °C for 10 min, separated by sodium dodecyl sulfate–polyacrylamide gel electrophoresis on a 10% acrylamide gel, and transferred onto a nitrocellulose membrane. The blot was blocked with 5% nonfat milk, and probed with primary antibodies against DRA (mouse monoclonal, 1:500, sc-376187; Santa Cruz, Dallas, TX, USA), CFTR antibody 217 (1:200 dilution; the University of North Carolina at Chapel Hill, Chapel Hill, NC, USA and Cystic Fibrosis Foundation Therapeutics, Inc.), glyceraldehyde-3-phosphate dehydrogenase (mouse monoclonal, 1:5000, G8795; Sigma-Aldrich), β-actin (mouse monoclonal, 1:5000, A2228; Sigma-Aldrich) overnight at 4 °C, followed by secondary antibody against mouse IgG (1:10,000) for 1 h at room temperature. Protein bands were visualized and quantitated using an Odyssey system and Image Studio Lite Ver 4.0 (LI-COR Biosciences, Lincoln, NE, USA).

### 4.5. Surface Biotinylation

At 4 °C, cells were incubated with 1.5 mg/mL N-hydroxysulfosuccinimide (NHS)-SS-biotin N-Hydroxysulfosuccinimide- and solubilized by lysis buffer. A small proportion (50 µg) of the protein lysate was collected as the total lysate, while the rest was incubated with avidin-agarose beads overnight. The beads were centrifuged and washed with lysis buffer containing 0.1% Triton X-100. Biotinylated proteins were eluted from the beads and collected as the surface fraction. Immunoblotting was performed as described earlier and the percentage of surface expression of DRA was calculated as previously reported by loading equal volumes for each total and surface fraction [17]. The surface-to-total ratios were calculated separately for the DRA upper band (highly glycosylated, ~102 kilodaltons in size), the lower band (core glycosylated, ~85 kilodaltons in size), as well as for both bands together.

### 4.6. Measurement of Cl^−^/HCO_3_^−^ Exchange Activity

A detailed procedure for measuring Cl^−^/HCO_3_^−^ exchange activity in Caco-2/BBe monolayers has been described previously [17]. In brief, the activity was fluorometrically measured by using the pHi-sensitive dye BCECF-AM in a customized chamber allowing simultaneous but separate apical and basolateral superfusion. The monolayers were rinsed and equilibrated in sodium solution (138 mM NaCl, 5 mM KCl, 2 mM CaCl2, 1 mM MgSO4, 1 mM NaH2PO4, 10 mM glucose, 20 mM HEPES, pH 7.4) for 60 min at 37 °C, then loaded with BCECF-AM (10 mM) in the same solution for another 30 min, and mounted in a fluorometer (Photon Technology International, Birmingham, NJ, USA). The basolateral surface was superfused continuously with Cl- solution (110 mM NaCl, 5 mM KCl, 1 mM CaCl_2_, 1 mM MgSO_4_, 10 mM glucose, 25 mM NaHCO_3_, 1 mM amiloride, 5 mM HEPES, 95% O_2_/5% CO_2_), while the apical side was superfused with a shift between Cl- solution or Cl^−^ -free solution (110 mM Na-gluconate, 5 mM K-gluconate,5 mM Ca-gluconate, 1 mM Mg-gluconate, 10 mM glucose, 25 mM NaHCO_3_, 1 mM amiloride, 5 mM HEPES, 95% O_2_/5% CO_2_). The switch between Cl^−^ solution and Cl^−^ -free solution results in HCO_3_^−^ uptake across the apical membrane mediated by Cl^−^/HCO_3_^−^ exchangers (such as DRA). The subsequent change of intracellular pH was recorded and the rate of initial alkalization was calculated using Origin 8.0 (Origin-Lab., Northampton, MA, USA) as an indicator of Cl^−^/HCO_3_^−^ exchange activity. The specificity of the contribution of DRA to the rate of alkalinization was determined during standardization by determining the prevention of alkalinization by the apical presence of the DRA inhibitor Ao250 (10 µM) (gift of A. Verkman).

### 4.7. Detergent-Soluble (DS) and -Insoluble (DI) Fractions of Total Membranes

Separation of the total membrane preparation into detergent-soluble and detergent-insoluble fractions was as we reported previously [7,47]. Post-confluent Caco-2/BBe cells in submerged culture or after ALI modifications were homogenized by passing them through a 1-mL syringe/26-gauge needle in TNE buffer A containing 25 mM Tris, pH 7.4, 150 mM NaCl, 50 mM NaF, 5 mM EDTA, 1 mM Na_3_VO_4_, and protease inhibitors. Nuclei and debris were removed by centrifugation at 3000× *g* for 15 min at 4 °C. The total membranes were pelleted by ultracentrifugation at 100,000× *g* for 30 min at 4 °C. Total membranes were then solubilized with cold buffer A supplemented with 0.5% Triton X-100 and then incubated at 4 °C for 30 min on a rotary shaker and subjected to ultracentrifugation at 100,000× *g* for 30 min at 4 °C. The supernatant is referred to as the DS fraction. The pellet was resuspended in 1× SDS-PAGE loading buffer (equal volume to the DS fraction) as the DI fraction.

### 4.8. Reverse Transcription and Real-Time PCR

A reported procedure to measure the transcriptional expression of target genes using primer pairs was followed [18,44]. In brief, the PureLink RNA Mini Kit (Life Technologies, CA, USA) was used to extract total RNA from Caco-2/BBe cells or colonoid monolayers. Then, the complementary DNA was synthesized from the extracted RNA using SuperScript VILO Master Mix (Life Technologies). Quantitative real-time PCR (qRT-PCR) was performed on a QuantStudio 12K Flex real-time PCR system (Applied Biosystems, Foster City, CA, USA) by using Power SYBR Green Master Mix (Life Technologies, CA, USA). Samples were run in triplicate, and the relative mRNA expression level of targeted genes was calculated from the 2-ΔΔCT method normalized with the vehicle control and housekeeping β-actin RNA.

### 4.9. Measurement of Active Anion Secretion (Short-Circuit Current, Isc)

The short-circuit current as an indicator of active electrogenic anion secretion in Caco-2/BBe cells was determined by the Ussing chamber/Voltage Clamp Technique [17]. In brief, Caco-2/BBe monolayers were mounted in Ussing chambers and incubated in Krebs-Ringer bicarbonate (KBR) buffer (115 mM NaCl, 25 mM NaHCO_3_, 0.4 mM KH_2_PO_4_, 2.4 mM K_2_HPO_4_, 1.2 mM CaCl_2_, 1.2 mM MgCl_2_, pH 7.4) continuously gassed with 95% O_2_/5% CO_2_ at 37 °C and connected to a voltage-current clamp apparatus (Physiological Instruments) via Ag/AgCl electrodes and 3 M KCl agar bridges. 10 mM glucose was supplemented as an energy substrate on the basolateral side, while 10 mM mannitol was added to the apical chamber to maintain the osmotic balance. Current clamping was employed and short-circuit current was recorded every 1 or 5 s by the Acquire and Analyze software 2.2.2 (Physiological Instruments, San Diego, CA, USA). 10 μM forskolin was added to the apical and basal chamber to initiate cAMP-stimulated anion secretion.

### 4.10. Statistics

GraphPad Prism (version 6.01, GraphPad Software, San Diego, CA, USA) was used to perform the statistical analysis. Data are mean ± s.e.m. of at least three independent experiments, with an error bar equaling one s.e.m. A two-tail Student’s *t*-test was used for statistical comparison between two groups, while a one-way analysis of variance (ANOVA) followed by post hoc Turkey was adopted if more than two different groups were compared. *p* < 0.05 was considered as the threshold of statistical significance.

## Figures and Tables

**Figure 1 ijms-24-08273-f001:**
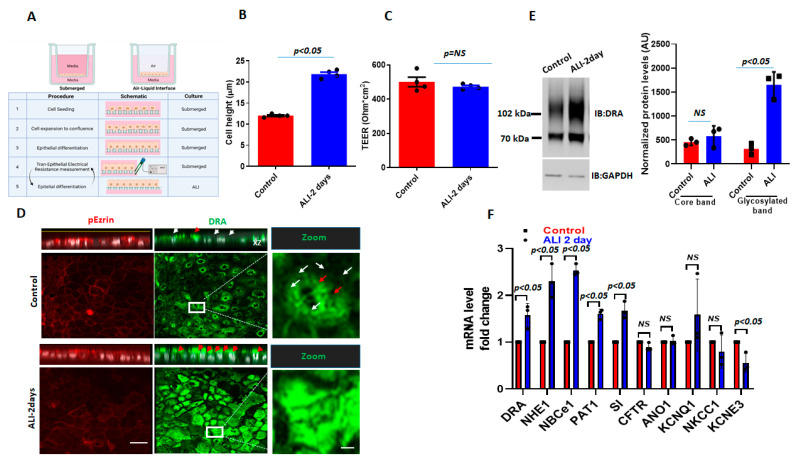
ALI culture increases DRA mRNA and protein expression in Caco-2/BBe cells. (**A**) A schematic overview of the procedures for creating an ALI culture in intestinal epithelial cells, including TEER measurements. (**B**) Cell height in Caco-2/BBe cells grown as a submerged culture for 12–14 days (red) or followed by 2-day ALI treatment after 12 days post-confluency (blue). (**C**) TEER measurements of Caco-/2BBe monolayers from submerged and ALI cultures. Mean ± SEM of *n* = 3–4 independent experiments shown in B and C. *p*-values represent unpaired Student’s *t*-test. (**D**) Confocal fluorescence microscopy of DRA (green), p-Ezrin (red), and nuclei (white). XZ orthogonal views (upper) and XY projections at the level of the apical membrane (lower). White arrowheads represent small aggregates of DRA-containing structures and red arrowheads represent large aggregates (platforms) of DRA on the apical side of Caco-2/BBe cells. Inserts show expanded areas. Scale bar: 10 µm; insert: 5 µm. (**E**) Left: IB of DRA and GAPDH of submerged (control) and ALI cultures. Right: densitometric analysis of DRA blots from *n* = 3 independent experiments separately quantitating DRA upper and lower bands normalized to GAPDH. Results are Means ± SEM. (**F**) Quantitative real-time PCR analysis of changes in mRNA expression of multiple ion-transport proteins and sucrase-isomaltase. Results are shown as Mean ± SEM of fold change. *n* = 3–5 independent experiments; NS = nonsignificant; *p*-values represent unpaired Student’s *t*-test.

**Figure 2 ijms-24-08273-f002:**
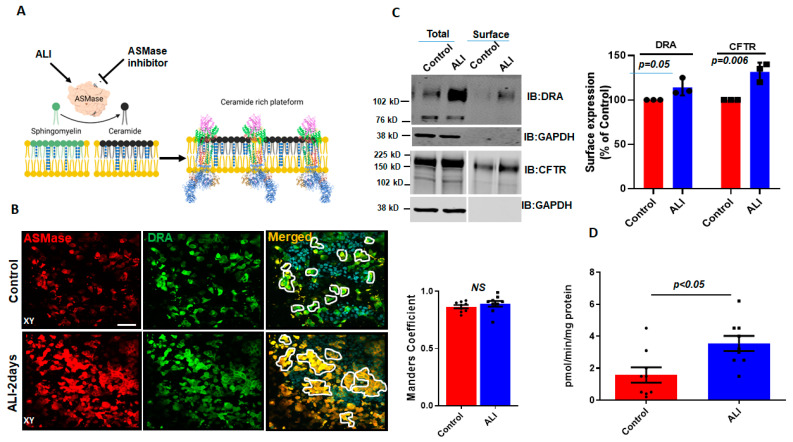
Increase in surface expression of DRA and ASMase activity in response to ALI. (**A**) Cartoon showing the role of ASMase in changing membrane ceramide amount and fusion of lipid rafts aggregates due to the increase in membrane ceramide content. (**B**) Immunofluorescence detection of ASMase (red) and Flag-DRA (green) in the apical membrane of unpermeabilized, well-differentiated Caco-2/BBe cells grown as submerged cultures or followed by 2-day ALI modifications 12–14 days post-confluency. Co-localization of endogenous ASMase and Flag-DRA detected in cells grown as submerged cultures and enhanced by 2-day ALI modifications. A single plane (XY) at the apical surface of the Z-stack section is shown. Manders’ overlap coefficient showing 90% spatial colocalization of Flag-DRA and ASMase inside enclosed areas. Scale bar: 50 µm, *n* = 3 repetitions of the experiment were performed. (**C**) Left: representative immunoblot and densitometric analysis of total and cell surface biotinylation of DRA (above) and CFTR (below) in Caco-2/BBe cells grown as submerged or 2-day ALI modifications after 12–14 days post-confluency. Right: quantitation of surface-to-total ratio normalized to GAPDH and expressed as a percent of control. Results are Mean ± SEM, *n* = 3 independent experiments. (**D**) Normalized ASMase activity in 12–14 days post-confluent Caco-2/BBE cells and changes in activity after 2-day ALI modifications. Results are Mean ± SEM, *n* = 3–8 independent experiments. NS = nonsignificant; *p*-values represent unpaired Student’s *t*-test.

**Figure 3 ijms-24-08273-f003:**
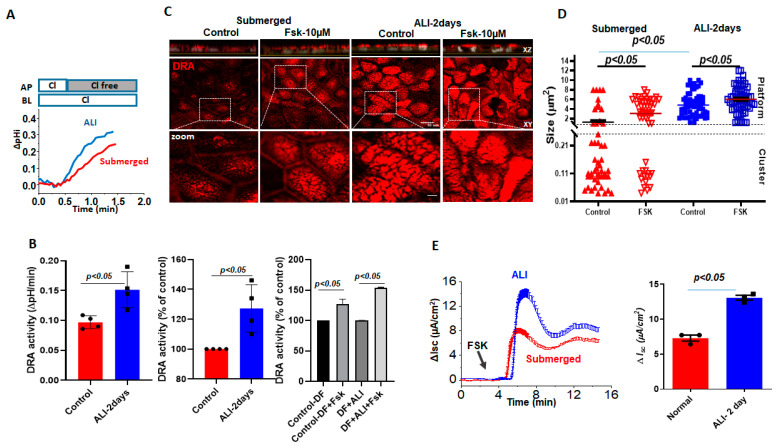
Increase in DRA basal activity and agonist stimulation in Caco-2/BBe cells following ALI exposure. Difference in basal and stimulated DRA activity and expression between Caco-2BBe cells grown as a submerged culture for 12–14 days (red) or followed by 2-day ALI (blue) cultures 12 days post-confluency: (**A**) representative trace and (**B**) basal Cl^−^/HCO_3_^−^ exchange activity and forskolin stimulation. Left: initial rates as ΔpH/min; center and right normalized as percent of control; results are Mean ± SEM, *n* = 3 independent experiments. *p*-values represent unpaired Student’s *t*-test. (**C**) Left: representative confocal image showing changes in sizes of endogenous DRA-containing clusters and appearance of the platform (clusters > 2 µm) comparing submerged and ALI-exposure cultures prior to and following forskolin (10 µM, 10 min) stimulations. All specimens were immunostained for DRA (red) and nuclei (gray) with XZ sections (above), XY sections at the level of BB (middle), and a zoomed-in view of the boxed area below. Scale bars: 10 µm; insert 5 µm. (**D**) Quantitation of DRA-containing areas as in (**C**). DRA-containing areas with dimensions close to the limit of optical resolution (<0.25 µm) were considered clusters, whereas large aggregates (≥2 µm dia.) were considered platforms. Size of aggregates in µm calculated for ~100 individual clusters and platforms from three areas/per monolayer/condition. Results are Mean ± SEM, *n* = 3–8 independent experiments. *p*-values represent unpaired Student’s *t*-test. (**E**) A representative trace (left) and summary (right) of ΔI_sc_ responses showing forskolin-induced Δi_sc_ from submerged culture and 2-day-exposed ALI monolayers following 12–14 days of confluency. Results are Mean ± SEM, *n* = 8–12 filters/conditions from 3 independent experiments. *p*-values represent unpaired Student’s *t*-test.

**Figure 4 ijms-24-08273-f004:**
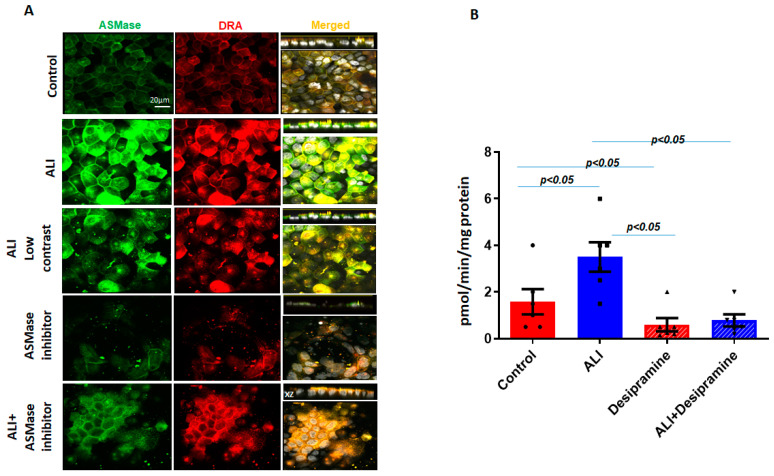
Increase in surface DRA expression due to ALI blocked by the ASMase inhibitor, desipramine. The effect of the ASMase inhibitor, desipramine, was determined by the apical plasma membrane localization of DRA in Caco-2/BBe cells grown as a submerged culture for 12 days or followed by ALI 2-day cultures. Flag-DRA was transduced into Caco-2/BBe cells, 24 h before ALI initiation. Transduced cells were exposed to ALI 2-day cultures or were pretreated with 13 µM of desipramine 1 hr before ALI was initiated and during ALI treatment. (**A**) A representative confocal image after immunostaining for Flag-DRA (red), ASMase (green), and nuclei (white). A single plane of a multi-Z-stack is shown. All panels represent the XY projection near the top of the cell; the right panel includes the XZ projection on the top. Scale bar 20 μm. (**B**) Normalized ASMase activity in Caco-2/BBe cells expressing Flag-DRA shown for four conditions in A. Pretreating cells with 13 µM of desipramine reduces ASMase activity below the control level, suggesting there is some ASMase activity under basal conditions. Results are Mean ± SEM; *n* = 3–8 independent experiments. Statistical analysis was performed using ANOVA.

**Figure 5 ijms-24-08273-f005:**
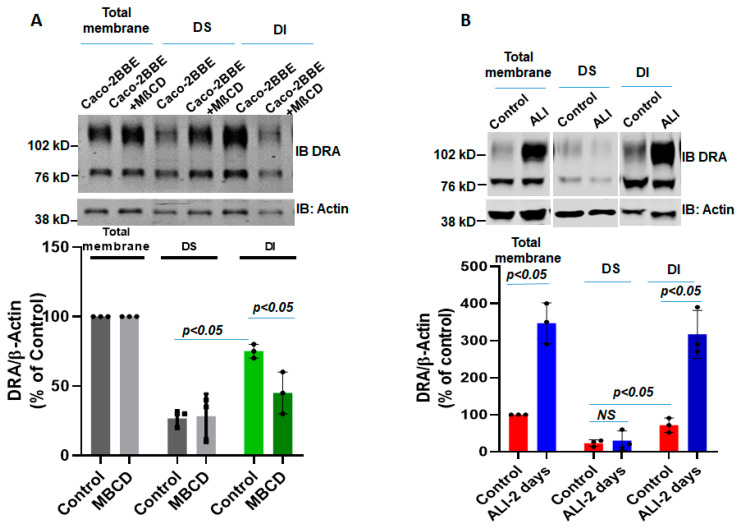
ALI causes an increase in the detergent-insoluble fraction of DRA in intestinal epithelial cells. DRA is primarily associated with the detergent-insoluble (DI) fractions of Caco-2/BBe cells and ALI 2-day exposure causes an increase in the DI fraction of DRA of total membranes. (**A**) DI vs. DS distribution of DRA in the presence or absence of methyl-β-cyclodextrin (MβCD; 10 µM, 1 h). The single experiment mentioned above and Mean ± SEM of *n* = 3 similar experiments are shown below. (**B**) Total membranes from submerged and ALI 2-day-exposed cells were solubilized in a buffer containing Triton X-100 and then detergent-soluble (DS) and DI fractions were isolated, as described in the Materials and Methods Section. Equal amounts of proteins (∼50 μg) from DI and DS fractions were separated on 10% PAGE and then analyzed by Western blotting for DRA and actin expression. The data were quantified by densitometric analysis and expressed as a percent of control and fold change representing Means ± SEM of 4 determinations. NS = nonsignificant; *p*-values represent unpaired Student’s *t*-test.

**Figure 6 ijms-24-08273-f006:**
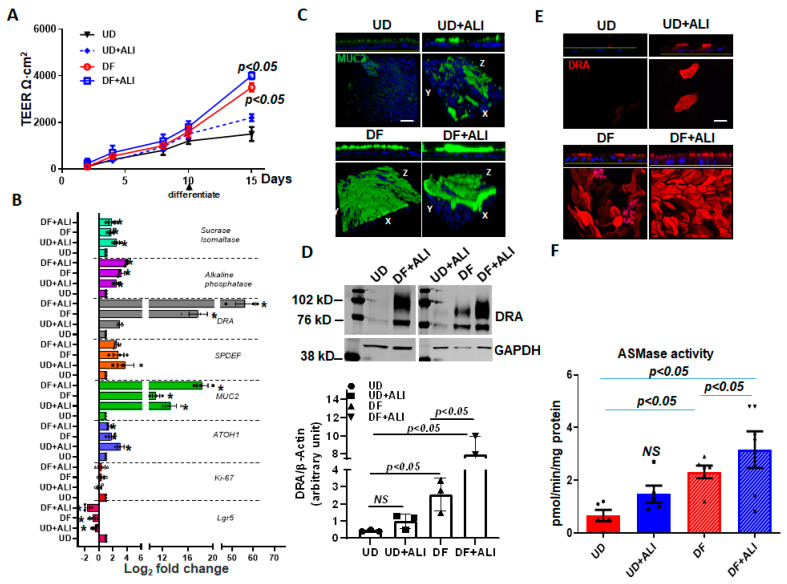
ALI increased differentiation of human colonoids. Human intestinal colonic organoids (colonoids) derived from biopsies were grown as confluent epithelial monolayers on permeable inserts. Post-confluency monolayers studied in all these experiments and compared: (1) UD; (2) UD + ALI; (3) 5-day DF; (4) 5-day DF + ALI. (**A**) Changes in TEER in response to differentiation or ALI modifications are shown. Results are Means ± SEM of *n* = 3–4 experiments. *p*-value compared with the UD control. (**B**) Relative mRNA levels of genes used to evaluate proliferation and differentiation, including DRA by qRT-PCR. Messenger RNA levels are normalized to 18S ribosomal RNA expression. Results are normalized to UD set as 1 and expressed as Log_2_ fold change. Data were analyzed using a two-tailed Student’s *t*-test with Welch’s correction. * *p* < 0.05 compared with UD control. Results are Means ± SEM of *n* = 3–4 independent experiments. Atoh, atonal homolog 1; Lgr5, leucine-rich repeat-containing G protein-coupled receptor; Muc2, mucin 2; ki67, nuclear protein ki67; SPDEF, SAM pointed domain containing ETS transcription factor. (**C**) Methanol–Carnoy’s fixed colonoid monolayers stained with anti-MUC2 (green) and nucleus stained with DAPI (blue). Representative confocal XZ (above) and 3D-XYZ (below) projections depicting the MUC2 layer in colonoid monolayers are shown. Scale bar 10 μm. (**D**) Representative Western blot and densitometry analyses from multiple experiments show changes in DRA protein expression. Results are Means ± SEM of *n* = 3 experiments. (**E**) Confocal fluorescence microscopy of DRA (red) expression in colonoids. Upper: XZ projection; lower: XY projection at the level of the apical domain. Scale bar 10 μm. (**F**) Normalized ASMase activity in colonoids. Results are means ± SEM of 3–5 independent experiments. NS = nonsignificant; *p*-value is compared with the UD control. *p*-values represent unpaired Student’s *t*-test.

**Figure 7 ijms-24-08273-f007:**
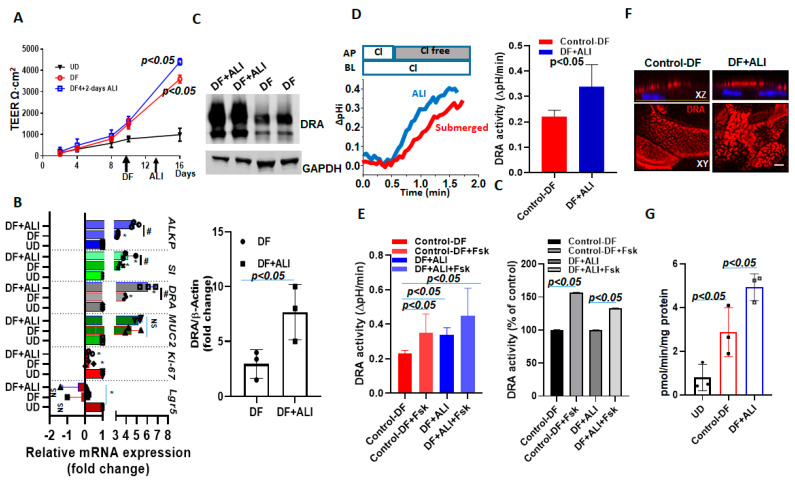
DRA expression and activity were further enhanced by exposing differentiated human colonoids to ALI culture. Human colonoids were grown as confluent epithelial monolayers on permeable inserts. Post-confluency monolayers were exposed to either differentiation medium (DF) for 6 days or to 4 days DF followed by 2-day ALI modifications. (**A**) Changes in TEER in response to differentiation or ALI modification are shown. Points are Means ± SEM, *n* = 4. (**B**) Relative mRNA expression of DRA by quantitative PCR. Messenger RNA levels were normalized to 18S ribosomal RNA expression. The results are normalized to UD set as 1 and expressed as fold change. Results are Mean ± SEM from *n* = 3 independent experiments. NS=nonsignificant; * *p* < 0.05 vs. UD control, ^#^ *p* < 0.05 vs. 6-day DF control. *p*-values were analyzed using two-tailed Student’s *t*-test with Welch’s correction. (**C**) A representative Western blot (above) and densitometry analysis Means ± SEM (below) from multiple experiments (*n* = 3) showing an increase in DRA protein expression in 4-day DF + 2-day ALI compared to 6-day DF condition. *p*-values represent unpaired Student’s *t*-test. (**D**) A representative trace of basal Cl^−^/HCO_3_^−^ exchange activity (left) and Means ± SEM of *n* = 3 independent experiments (right) and (**E**) basal DRA activity and forskolin (10 µM, 10 min) stimulation comparing 6-day DF and 4-day DF + 2-day ALI conditions, shown as DRA activity and as a percent of control, *n* = 3. Statistical analysis was performed using ANOVA. (**F**) Confocal fluorescence microscopy of DRA (red) and nucleus stained with DAPI (blue) expression in colonoids; upper XZ and lower XY project at the level of the apical domain. Scale bar: 10 μm. (**G**) Normalized ASMase activity in colonoids comparing UD, 6-day DF, and 4-day DF + 2-day ALI. Results are Means ± SEM, *n* = 3 independent experiments. *p*-values represent unpaired Student’s *t*-test.

## Data Availability

Johns Hopkins Medicine IRB committee.

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
