# Peer review of "The Air–Liquid Interface Reorganizes Membrane Lipids and Enhances the Recruitment of Slc26a3 to Lipid-Rich Domains in Human Colonoid Monolayers"

_ijms, 2023, doi:10.3390/ijms24098273_

Round 1

Reviewer 1 Report

The manuscript entitled “Air-Liquid-Interface Reorganizes Membrane Lipid and Enhance Recruitment 1 of Slc26a3 to Lipid-Rich Domains in Human Colonoid Monolayers” by Ming Tse et al. demonstrates the formation of large membrane domains that contain acid sphingomyelinase and serve to cluster the intestinal Cl-/HCO3- antiporter Down Regulated in Adenoma (DRA). The activity and surface expression of this transporter depends on the integrity of ceramide-enriched domains and the acid sphingomyelinase as nicely demonstrated in studies using the functional acid sphingomyelinase inhibitor desipramine.

The manuscript is novel, well performed and very interesting. I have only a few comments:

1.) Fig. 4 is accurate, since it shows the acid sphingomyelinase surface expression at the same intensities and settings that after application of the inhibitor. However, this results in over-exposure and the authors should add a subpanel with optimal exposure time to see the distribution of the acid sphingomyelinase upon culture in ALI.

2.) The quantitative analysis of DRA in Fig. 5 should not be given in % of control. It would be better to show the real values, if possible.

3.) The authors may want to test whether the observed platforms are really formed by ceramide. This can be done using a monoclonal anti-ceramide antibody and confocal microscopy. However, since this is not a critical experiment, it might be also possible to discuss the role of ceramide in membrane domains and for the formation of membrane domains.

4.) The authors may want to check the manuscript carefully for typos.

Author Response

We thank the reviewers for supporting our manuscript. Based on minor comments from reviewers we have modified the manuscript and are submitting the modified manuscript. Please find the point-by-point response to the reviewers’ concerns below:

Response to Reviewer 1

The manuscript entitled “Air-Liquid-Interface Reorganizes Membrane Lipid and Enhance Recruitment 1 of Slc26a3 to Lipid-Rich Domains in Human Colonoid Monolayers” by Ming Tse et al. demonstrates the formation of large membrane domains that contain acid sphingomyelinase and serve to cluster the intestinal Cl-/HCO3- antiporter Down Regulated in Adenoma (DRA). The activity and surface expression of this transporter depends on the integrity of ceramide-enriched domains and the acid sphingomyelinase as nicely demonstrated in studies using the functional acid sphingomyelinase inhibitor desipramine.

The manuscript is novel, well performed and very interesting. I have only a few comments:

1.) Fig. 4 is accurate, since it shows the acid sphingomyelinase surface expression at the same intensities and settings that after application of the inhibitor. However, this results in over-exposure and the authors should add a subpanel with optimal exposure time to see the distribution of the acid sphingomyelinase upon culture in ALI.

Thanks for bringing up this point. We have replaced the ALI figure with a new one where we could lower the contrast and maintain some of the ASMase distribution. A subpanel has been added to figure 4A (ALI low contrast).

2.) The quantitative analysis of DRA in Fig. 5 should not be given in % of control. It would be better to show the real values, if possible.

Due to the huge variations in the arbitrary number between different experiments, we decided to express results as % of control. We also believe that % of control gives a better idea of change as it is expressed with respect to control.   

3.) The authors may want to test whether the observed platforms are really formed by ceramide. This can be done using a monoclonal anti-ceramide antibody and confocal microscopy. However, since this is not a critical experiment, it might be also possible to discuss the role of ceramide in membrane domains and for the formation of membrane domains.

We thank the reviewer for bringing up this point. We are currently performing similar experiments as understanding the role of ceramide in real time is the future direction of our project.

4.) The authors may want to check the manuscript carefully for typos.

Sorry about the typos. We fixed multiple typos in our previous manuscript.

Reviewer 2 Report

The manuscript by Tse et al. deals with membrane lipid reorganization by air liquid interface (ALI) in human colonoid monolayers. Down Regulated in Adenoma (DRA), the intestinal Cl- /HCO3 - antiporter is enriched in lipid rafts. They created ALI by removing apical media for Caco34 2/BBe cells or colonoid monolayer grown as submerged cultures. The authors described, that DRA expression and activity were increased in Caco-2/BBe cells and human colonoids using this method. ALI induced an increase in acid sphingomyelinase (ASMase) activity and ALI cultures expressed a larger number of DRA-containing platforms. Despiramine, an ASMase inhibitor, disrupted CRPs and decreased the ALI-induced increase in DRA expression. ALI enhanced DRA activity and expression by increasing the ASMase activity and platform formation in Caco-2/BBe cells and by increasing the differentiation of colonoids.  The manuscript will certainly be interest to other scientists in the field. This is a well-documented study and contains several cited paper which adequately supporting the major conclusions in the paper. The figures are well designed and illustrate the described information. However, I have outlined my concerns as follows:

1.           There are complete introduction parts in almost all paragraphs of Results section. For example in 140-147 lines. Please insert these informations into the Introduction or if it is needed into the Discussion section, and do not use citations in Results section.

2.           In 180-184 lines: These results are results of another research groups. As I have mentioned above please insert these informations into the Introduction or into the Discussion section, and do not use citations in Results section.

3. In 207-210 lines: Please described these ionformations in the Introduction section.

Author Response

We thank the reviewers for supporting our manuscript. Based on minor comments from reviewers we have modified the manuscript and are submitting the modified manuscript. Please find the point-by-point response to the reviewers’ concerns below:

Response to reviewer 2:

The manuscript by Tse et al. deals with membrane lipid reorganization by air liquid interface (ALI) in human colonoid monolayers. Down Regulated in Adenoma (DRA), the intestinal Cl- /HCO3 - antiporter is enriched in lipid rafts. They created ALI by removing apical media for Caco34 2/BBe cells or colonoid monolayer grown as submerged cultures. The authors described, that DRA expression and activity were increased in Caco-2/BBe cells and human colonoids using this method. ALI induced an increase in acid sphingomyelinase (ASMase) activity and ALI cultures expressed a larger number of DRA-containing platforms. Despiramine, an ASMase inhibitor, disrupted CRPs and decreased the ALI-induced increase in DRA expression. ALI enhanced DRA activity and expression by increasing the ASMase activity and platform formation in Caco-2/BBe cells and by increasing the differentiation of colonoids.  The manuscript will certainly be interest to other scientists in the field. This is a well-documented study and contains several cited paper which adequately supporting the major conclusions in the paper. The figures are well designed and illustrate the described information. However, I have outlined my concerns as follows:

  1. There are complete introduction parts in almost all paragraphs of Results section. For example in 140-147 lines. Please insert these informations into the Introduction or if it is needed into the Discussion section, and do not use citations in Results section.

We believe citing other references helps to give a quick overview of why the experiment is performed or how it was performed previously and is important information for the reader. Although now we only have a few which are very important. This is just a personal style of writing.

  1. In 180-184 lines: These results are results of other research groups. As I have mentioned above please insert these informations into the Introduction or into the Discussion section, and do not use citations in Results section.

Removed.

  1. In 207-210 lines: Please described these ionformations in the Introduction section.

We have now added this information in the introduction.

Thanks